# Cross-Sectional Association between Perceived Physical Literacy and Mediterranean Dietary Patterns in Adolescents: The EHDLA Study

**DOI:** 10.3390/nu15204400

**Published:** 2023-10-17

**Authors:** Gabriel Domínguez-Martín, Pedro J. Tárraga-López, José Francisco López-Gil

**Affiliations:** 1Consejería de Educación, Region of Murcia, 30071 Murcia, Spain; gabrieldomar@gmail.com; 2Departamento de Ciencias Médicas, Facultad de Medicina, Universidad de Castilla-La Mancha, 02006 Albacete, Spain; pjtarraga@sescam.jccm.es; 3One Health Research Group, Universidad de Las Américas, Quito 170124, Ecuador

**Keywords:** physical education, healthy eating, lifestyle, Mediterranean diet, teenagers, youths

## Abstract

Purpose: The objective of the current study was to analyze the relationship between perceived physical literacy (PPL) and adherence to the Mediterranean Diet (MedDiet) and its specific components in a sample of Spanish adolescents. Methods: In this cross-sectional study, we examined a sample of 845 adolescents (58.1% boys) aged 12–17 years from the *Valle de Ricote* (Region of Murcia, Spain). PPL was assessed by the Spanish Perceived Physical Literacy Instrument for adolescents (S-PPLI). To assess adherence to the MedDiet, the Mediterranean Diet Quality Index in children and adolescents (KIDMED) was applied. Results: In unadjusted analyses, adolescents with medium or high PPL were more likely to have optimal adherence to the MedDiet (medium PPL: OR = 1.69, 95% CI 1.20–2.40, *p*-adjusted = 0.003; high PPL: OR = 2.90, 95% CI 2.03–4.17, *p*-adjusted < 0.001). These findings remained significant after adjusting for sex, age, socioeconomic status, body mass index, energy intake, overall sleep duration, physical activity, and sedentary behavior (medium PPL: OR = 1.51, 95% CI 1.05–2.19, *p*-adjusted = 0.003; high PPL: OR = 2.27, 95% CI 1.53–3.39, *p*-adjusted < 0.001). Conclusions: PPL could play a relevant role in the adoption of healthy eating habits among adolescents. Adolescents with low or medium PPL were less likely to report optimal adherence to the MedDiet. Adolescents with high PPL seem to consume more fruits, vegetables, fish, pulses, nuts, and dairies (for breakfast). Conversely, these adolescents tend to skip breakfast less, as well as to consume less commercially baked goods or pastries at this meal.

## 1. Introduction

Physical literacy refers to the inclination, self-assurance, physical proficiency, awareness, and comprehension that people cultivate to sustain a suitable level of physical engagement throughout their lifetime [1,2]. Similar to how reading, writing, listening, and speaking come together to establish lifelong language skills, physical literacy is an ongoing process where various elements (such as physical ability, daily habits, comprehension, motivation, and self-assurance) synergistically interact to enable a lifetime of involvement and pleasure in physical pursuits [3]. Physical literacy goes beyond the scope of traditional physical education in schools or organized sports, presenting a more inclusive notion of physical engagement that is not tied to one’s skill level. By employing teaching methods and embracing fresh perspectives, physical literacy holds the potential to introduce more practical models of physical proficiency and activity that cater to a broader populace. This opens doors for all individuals to engage as enthusiastic and active participants [4]. Physical literacy is cultivated by engaging in physical activities, representing a unique form of intelligence associated with the capacity to execute movements. This form of intelligence is distinct from mere physical activity; instead, it serves as a significant precursor to engaging in physical activities [5]. Adolescents have the right to receive a valuable educational encounter in high-quality physical education, which enhances their physical literacy and nurtures the subsequent essential characteristics: (a) their self-perception and self-assurance in a physical context; (b) their drive to participate actively; (c) their interactions with their surroundings, self-expression, and communication with peers; and (d) their comprehension and awareness of how to sustain physical activity [1,2]. Consequently, as adolescents enhance their physical literacy, they gain self-assurance in basic motor skills, coordination, and mastery within a dynamic setting. They are also capable of displaying both spoken and unspoken communication for engaging with peers within a physical context, and they find pleasure in exploring novel physical pursuits.

Conversely, the Mediterranean diet (MedDiet), a predominantly plant-focused dietary regimen that is gaining global popularity, is regarded as one of the most beneficial diet patterns for health [6]. Studies indicate that people can experience numerous advantages by integrating components of this diet into their dietary practices [7]. This dietary approach involves the substantial consumption of plant-derived foods (such as vegetables, fruits, grains, legumes, seeds, nuts, and potatoes); utilization of fresh, minimally processed, locally sourced foods in accordance with seasons; primary reliance on olive oil for fats; and moderate consumption of dairies (primarily yogurt and cheese), among other constituents [6]. Furthermore, the MedDiet has been suggested as a benchmark dietary model owing to its myriad health and nutritional advantages, its promotion of biodiversity, its significant sociocultural significance as a culinary tradition, its minimal environmental footprint, and its favorable economic contributions to local societies [8].

To our knowledge, no previous study has examined the relationship between physical literacy and eating healthy habits (e.g., adherence to the MedDiet). However, physical activity (which is closely related to physical literacy [5]) has also been associated with adherence to the MedDiet in the young population. In addition, healthy behaviors seem to cluster, as the scientific literature has suggested [9,10,11,12]. Therefore, an association between physical literacy and eating habits could not be ruled out. Due to the limited research on this particular relationship, gaining a more profound understanding of how physical literacy relates to adherence to the MedDiet could hold significant significance in formulating future intervention initiatives aimed at enhancing dietary behaviors among adolescents. Therefore, the objective of the current study was to analyze the relationship between physical literacy and adherence to the MedDiet and its specific components in a sample of Spanish adolescents.

## 2. Materials and Methods

### 2.1. Study Design and Population

This is a secondary cross-sectional study that used data from the Eating Healthy and Daily Life Activities (EHDLA) study. The comprehensive approach of the EHDLA study has been documented elsewhere [13]. The subjects involved in this research were adolescents from Spain, specifically aged between 12 and 17 years, and they attended three secondary schools located in the *Valle de Ricote* in the Region of Murcia. Data were collected during the 2021/2022 academic year. Of the initial 1378 adolescents (100.0%) from the EDHLA study, 383 (27.8%) were eliminated from the study due to insufficient information about their adherence to the MedDiet. Furthermore, additional participants were excluded because of incomplete data regarding PPL (*n* = 63; 4.6%), body mass index (*n* = 45; 3.3%), and energy intake (*n* = 42; 3.0%). Therefore, a sample of 845 adolescents (58.1% boys) was examined in this secondary cross-sectional study. To be involved in this study, parental or guardian consent was secured in written form for the chosen teenage participants. They were furnished with an informative document detailing the study’s objectives, as well as the assessments and surveys that would be administered. Additionally, the adolescents themselves were requested to provide their consent to take part.

This research adhered to the principles outlined in the Declaration of Helsinki and received approval from the Ethics Committee at the University of Murcia (ID 2218/2018), as well as from the Ethics Committee of the Albacete University Hospital Complex and the Albacete Integrated Care Management (ID 2021-85) for studies involving human participants.

### 2.2. Procedures

#### 2.2.1. Perceived Physical Literacy

Perceived physical literacy (PPL) was assessed by the Spanish Perceived Physical Literacy Instrument for adolescents (S-PPLI), which was previously validated for Spanish youth [14]. The Perceived Physical Literacy Instrument (PPLI) is an assessment tool with 18 items that was initially created for physical-education teachers [15]. The version aimed at adolescents consists of 9 items that are rated using a 5-point Likert scale, ranging from 1 (strongly disagree) to 5 (strongly agree). These 9 items of the PPLI are evenly distributed across three categories: (a) “knowledge and comprehension” (3 items), (b) “self-expression and interaction with others” (3 items), and (c) “self-perception and self-confidence” (3 items). For an additional analysis, we categorized the S-PPLI score into tertiles (after eliminating subjects with missing information) as follows: low PPL (9–31 points), medium PPL (32–36 points), and high PPL (31–45 points).

#### 2.2.2. Adherence to the Mediterranean Diet

Adherence to the MedDiet was assessed using the Mediterranean Diet Quality Index in children and adolescents (KIDMED) [16]. This index involves a 16-question test with a scoring range from −4 to 12 points. Negative scores are assigned to questions related to unhealthy aspects of the diet, while positive scores are given to questions about healthy aspects. The cumulative score is categorized as follows: high adherence (≥8 points), moderate adherence (4–7 points), and low adherence (≤3 points) to the MedDiet. For further analyses, we collapsed these categories into optimal adherence (≥8 points) or nonoptimal adherence (<8 points) to the MedDiet.

#### 2.2.3. Covariates

Adolescents provided self-reported information about their sex and age. The Family Affluence Scale (FAS-III) was used to assess their socioeconomic status, which involved summing up responses from six items related to their family’s possessions and amenities (such as bedrooms, vehicles, bathrooms, computers, travels, or dishwashers) [17]. The resulting FAS-III score ranged from 0 to 13, with greater scores indicating greater socioeconomic status. The body weight of the adolescents was measured using an electronic scale (with an accuracy of 0.1 kg) (Tanita BC-545, Tokyo, Japan), while height was determined by a portable height rod with an accuracy of 0.1 cm (Leicester Tanita HR 001, Tokyo, Japan). Subsequently, body mass index (BMI) was computed by taking the participants’ body weight in kilograms and dividing it by the square of their height in meters. Energy consumption was evaluated using a self-administered food frequency questionnaire that was previously validated for use within the Spanish population [18]. Overall sleep duration was determined by asking adolescents about their typical bedtime and wake-up time on both weekdays and weekends. The overall sleep duration was determined by determining the average sleep duration during both weekdays and weekends. This was achieved using the formula [(average sleep duration on weekdays × 5) + (average sleep duration on weekends × 2)] divided by 7. To evaluate physical activity and sedentary behavior among the adolescents, the Youth Activity Profile Physical (YAP) questionnaire was employed [19]. This self-administered questionnaire was previously validated and covered a 7-day period, with 15 different items categorized into sections such as out-of-school activities, school-related activities, and sedentary habits.

### 2.3. Statistical Analysis

For categorical variables, the descriptive data are presented as both the number (n) and the percentages (%) of observations in each category. For continuous variables, the descriptive data are presented as the median and interquartile range (IQR) of the values. Visual methods (i.e., density and quantile–quantile plots) and Shapiro–Wilk’s test were used to check normality of the variables. Associations between study variables and PPL status (tertiles) were tested using the Kruskal–Wallis *H* test (for continuous variables) or chi-square test (categorical variables). Furthermore, we used the chi-square test to examine the relationship between PPL status and adherence to the MedDiet and the different components of the KIDMED. Given that no interaction between sex and adherence to the MedDiet (*p* = 0.397) was found, we decided to examine both boys and girls together. To examine the association between the S-PPLI score and KIDMED score or adherence to the MedDiet among adolescents without any parametric assumptions on the nature of the relationship, generalized additive models (GAMs) were used. The GAM is a flexible model that can capture nonlinear connections in data without requiring a predefined mathematical structure. Restricted maximum likelihood (REML) for smoothness selection was applied [20], with a shrinkage approach employed as a function of thin plate regression spline smoothers [21]. To quantify the degree of nonlinearity of the curve, we used the effective degrees of freedom (*edf*) of GAM. Regarding PPL status (i.e., low PPL, medium PPL, and high PPL), an analysis of covariance was conducted to estimate the association with the KIDMED score (adjusting for several covariates), applying a nonparametric bias-corrected and accelerated (*BCa*) bootstrap method with 1000 samples. Likewise, to determine the odds ratio (OR) and the 95% confidence interval (CI) of the association between PPL status (i.e., low PPL, medium PPL, and high PPL) and adherence to the MedDiet (i.e., nonoptimal MedDiet or optimal MedDiet), we conducted binary logistic regression analyses. For both types of analyses, we applied a correction for multiple comparisons by using the false discovery rate *p*-value method developed by Benjamini and Hochberg [22]. Models were adjusted for sex, age, socioeconomic status, body mass index, energy intake, overall sleep duration, physical activity, and sedentary behavior. All statistical analyses were conducted with the statistical software R (Version 4.3.1) (R Core Team, Vienna, Austria) and RStudio (Version 2023.03.1) (Posit, Boston, MA, USA). Statistical significance was indicated by a *p*-value < 0.05.

## 3. Results

The main characteristics of the adolescents examined based on their PPL status are displayed in Table 1. The overall proportion of adolescents with optimal adherence to the MedDiet was 38.1%. In relation to PPL status, the highest KIDMED score and adherence to the MedDiet were observed in adolescents with high PPL (KIDMED score: median = 8.0; IQR = 3.0; adherence to the MedDiet: 51.4%). Conversely, the lowest KIDMED score and adherence to the MedDiet were found among those adolescents with low PPL (KIDMED score: median = 6.0; IQR = 4.0; adherence to the MedDiet: 26.7%).

Figure 1 displays the proportion of the different KIDMED items met according to PPL status. Additional information on this association can be found in Appendix A. Based on PPL, statistically significant differences were observed for 12 of the 16 items (*p* < 0.05 for all). Adolescents with high PPL showed a higher proportion of fruit, vegetable, fish, pulse, nut, and dairy consumption (for breakfast) in comparison with their counterparts with medium or low PPL (*p* < 0.05 for all). Conversely, adolescents with high PPL reported a lower proportion of skipping breakfast and commercially baked goods or pastries’ consumption (for breakfast) compared to those with medium or low PPL (*p* < 0.05 for both).

Figure 2 presents smoothed functions from GAMs for the KIDMED score and optimal adherence to the MedDiet as a function of the S-PPLI score. While the approximate significance of smooth terms was statistically significant for both outcomes (*p* < 0.001), upon examining the figure and *edf*, we observed a nonlinear relationship between the S-PPLI score and the KIDMED score (*F* = 3.31; *edf* = 2.82; *p* < 0.001). Furthermore, for the KIDMED score, we found a significant range between 13 and 31 points on the S-PPLI score, indicating that a score within this range was inversely associated with the KIDMED score. Another significant range was observed at >34 points on the S-PPLI, showing the opposite direction compared to the significant range between 13 and 31 points on the S-PPLI. This suggests that a score within this range was associated with the KIDMED score. Regarding adherence to the MedDiet, given that the *edf* that was obtained is close to 1, it is close to being a linear term (*χ*^2^ = 11.22; *edf* = 0.94; *p* < 0.001). Additionally, a significant range was found between 9 and 33 points on the S-PPLI, indicating that the score was inversely associated with optimal adherence to the MedDiet among participants within this range of values. Additionally, for participants with an S-PPLI score > 33 points, an opposite direction was observed compared to the significant range between 9 and 33 points on the S-PPLI. This indicates an association between the score and adherence to the MedDiet in adolescents within this range.

The estimated marginal means of KIDMED scores and their *BCa* bootstrapped 95% CI based on PPL status are shown in Figure 3. Adolescents with high PPL showed the greatest KIDMED score (mean = 7.1; *BCa* bootstrapped 95% CI 6.8–7.4), followed by those with medium PPL (mean = 6.8; *BCa* bootstrapped 95% CI 6.5–7.0) and those with low PPL (mean = 6.0; *BCa* bootstrapped 95% CI 5.7–6.3). Significant differences were found between low PPL and medium PPL, as well as between low PPL and high PPL (*p* = 0.003 for both).

Figure 4 depicts the unadjusted and adjusted odds ratio of having an optimal adherence to the MedDiet according to PPL status. In unadjusted analyses (Figure 4A), adolescents with medium or high PPL were more likely to have optimal adherence to the MedDiet (medium PPL: OR = 1.69, 95% CI 1.20–2.40, *p*-adjusted = 0.003; high PPL: OR = 2.90, 95% CI 2.03–4.17, *p*-adjusted < 0.001). These findings remained significant after adjusting for sex, age, socioeconomic status, body mass index, energy intake, overall sleep duration, physical activity, and sedentary behavior (medium PPL: OR = 1.51, 95% CI 1.05–2.19, *p*-adjusted = 0.043; high PPL: OR = 2.27, 95% CI 1.53–3.39, *p*-adjusted < 0.001) (Figure 4B), indicating a significant association between PPL and adherence among adolescents.

## 4. Discussion

In general, our results indicate that higher PPL is related to greater adherence to the MedDiet, with higher consumption of some specific Mediterranean dietary components (i.e., fruits, vegetables, fish, pulses, nuts, and dairies (for breakfast)), as well as with lower odds of skipping breakfast and consuming commercially baked goods or pastries for breakfast. There is limited empirical evidence linking PPL and health outcomes, but the literature supports an association between the greater physical domain of physical literacy and more desirable health outcomes. A review by Cornish et al. [23] found that physical literacy was associated with lower body mass index and body weight, waist circumference, cardiorespiratory fitness, physical activity, and sedentary behavior. In the same vein, a meta-analysis by Carl et al. [24] pointed out that, despite meaningful heterogeneity across the subgroups, physical literacy interventions increase physical competence, physical activity behavior, knowledge and understanding in relation to physical activity, overall physical literacy, motivation, and confidence. Although no previous studies have analyzed the specific relationship between physical literacy and adherence to the MedDiet, there are some possible mechanisms that could justify these findings.

First, one possible reason may lie in the tendency of healthy behaviors to cluster together. The scientific literature has shown how healthy behaviors (e.g., physical activity and diet) tend to cluster together among youths [9,10,11,12]. In this sense, Melby et al. [25] observed that a higher PPL was linked with sport and exercise participation among Danish adolescents. Similarly, an association between greater PPL and higher physical fitness (both objective-measured [26,27] and self-perceived [28,29]) has been reported. In addition, the relationship between PPL and physical fitness seems to be directly mediated by moderate-to-vigorous physical activity [26]. Moreover, one longitudinal study by García-Hermoso et al. [30] observed that physical education attendance (which is closely related to physical literacy) was associated with healthy behaviors (i.e., recommended physical activity, screen time, and sleep duration) in adolescence and that association was maintained in adulthood. Despite the lack of studies, higher physical literacy has been linked with certain health outcomes (e.g., physical activity and physical fitness), so it may also (at least in part) explain healthier eating behaviors.

Second, physical literacy involves setting and achieving physical goals (i.e., goal-oriented mindset), which could also explain these results. Young people who are proficient in physical activities might extend their goal-oriented mindset to their dietary choices, recognizing that the MedDiet can complement their physical goals and, thus, adhere to it more consistently [31,32,33]. For instance, Lirola et al. [31] showed that physical education classes can positively influence the adoption of healthy eating habits. Similarly, Depboylu and Kaner [32] observed that children who reported engaging in regular physical activity were more likely to have optimal adherence to the MedDiet. Furthermore, a previous study by Miguel-Berges et al. [33] found an association between meeting PA (and screen time) recommendations and healthier dietary patterns (i.e., fruit, vegetable, and water consumption) and lower consumption of energy-dense foods (i.e., salty snacks, desserts, sodas, and sweets).

Third, engaging in physical activities can influence appetite regulation. Adolescents who are physically active might develop a better sense of hunger and satiety cues, which can help them adhere to the portion control and balanced eating approach of the MedDiet. In this sense, desirable movement behavior profiles, which include physical activity, sedentary behaviors, and sleep, have been associated with better appetite control and improved eating habits [34], as well as optimal adherence to the MedDiet [11]. Exercise can affect appetite regulation in adolescents [35]. Acute exercise may suppress appetite by lowering concentrations of ghrelin and other appetite-regulating hormones [36]. Physical activity can positively change food consumption due to improved appetite control combined with a higher motivation to engage in healthy behavior [37].

Fourth, a higher level of physical literacy often includes knowledge about nutrition and its impact on overall health and performance. In this sense, a large study among Italian adults showed that nutrition knowledge is a driver of adherence to the MedDiet [38]. Therefore, adolescents who are physically literate are more likely to understand the benefits of a balanced diet such as the MedDiet, leading to better food choices. Supporting this idea, Keske et al. [39] intended to teach the topic of a healthy diet in a physical education class to students through physical literacy. The researchers observed that following the intervention, adolescents became more attentive to the nutritional components of their food and exhibited heightened awareness of their dietary choices. More specifically, adolescents who engaged in any sports showed higher nutrition literacy, food literacy, and healthy eating behaviors (i.e., the Adolescent Nutrition Literacy Scale, Interactive Nutrition Literacy, Critical Nutrition Literacy, and Adolescent Food Habit Checklist scores) [40]. Adolescents with healthy behaviors associated with overall well-being (i.e., sufficient sleep, regular physical activity, and proper diet) seem to easily distinguish between concepts such as healthy and unhealthy foods but face challenges in explaining terms such as energy-dense foods and processed foods [41]. In addition, a recent meta-analysis reported that lifestyle-based interventions (including physical activity) improve adherence to the MedDiet in children and adolescents [42].

Despite the findings obtained, this study must be interpreted while bearing in mind certain limitations. As a result of employing a cross-sectional approach in this study, it is not possible to establish a direct causal link based on the results. Additional research employing diverse methodologies, such as experimental approaches, is necessary to investigate whether heightened levels of PPL are associated with greater adherence to the MedDiet among adolescents. Nonetheless, while clinical trials are necessary to establish a direct cause-and-effect relationship between dietary patterns and sleep, as well as to uncover the underlying mechanisms, existing data suggest a reciprocal connection between these lifestyle factors [11,12,43]. Likewise, employing questionnaires for collecting data about PPL and MedDiet may introduce bias, as variations in willingness to share information or inaccuracies in recall details could influence the results. Conversely, one strength of this study is that, to date, this is the first study examining the relationship between PPL and adherence to the MedDiet among adolescents. Similarly, our analyses were adjusted for sociodemographic (i.e., age, sex, and socioeconomic status), anthropometric (i.e., body mass index), and lifestyle variables (i.e., energy intake, physical activity, sedentary behavior, and sleep duration), making our findings more robust.

## 5. Conclusions

PPL could play a relevant role in the adoption of healthy eating habits among adolescents. Adolescents with low or medium PPL were less likely to report optimal adherence to the MedDiet. Adolescents with high PPL seem to consume more fruits, vegetables, fish, pulses, nuts, and dairies (for breakfast). Conversely, these adolescents tend to skip breakfast less and consume less commercially baked goods or pastries at this meal. These findings could be used to consider physical literacy in interventions aimed at increasing adherence to healthy eating habits in adolescents.

## Figures and Tables

**Figure 1 nutrients-15-04400-f001:**
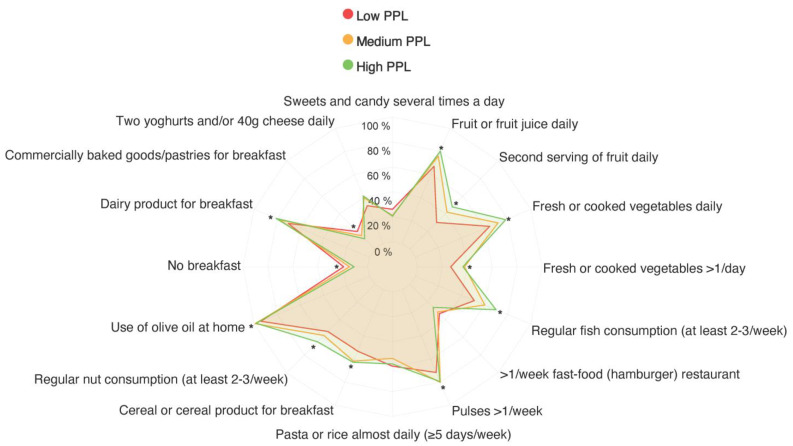
Mediterranean Diet Quality Index in children and adolescents according to physical literacy status among adolescents. Data are expressed as percentages. PPL, perceived physical literacy. * A *p*-value < 0.05.

**Figure 2 nutrients-15-04400-f002:**
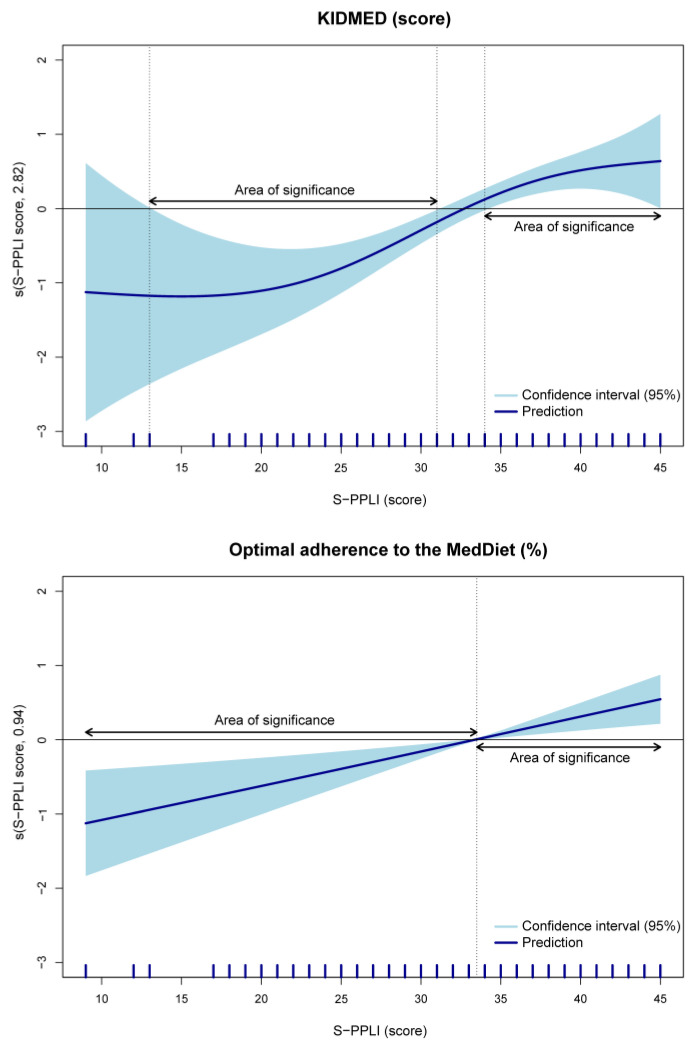
Association between perceived physical literacy and KIDMED score among adolescents, using generalized additive models. Adjusted for sex, age, socioeconomic status, body mass index, energy intake, overall sleep duration, physical activity, and sedentary behavior. KIDMED, Mediterranean Diet Quality Index in children and adolescents; MedDiet, Mediterranean diet; S-PPLI, Spanish Perceived Physical Literacy Instrument.

**Figure 3 nutrients-15-04400-f003:**
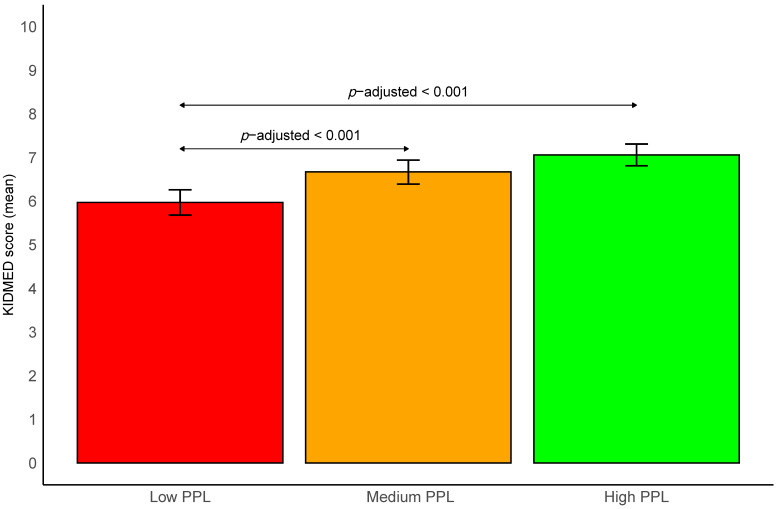
Estimated marginal means of Mediterranean Diet Quality Index in children and adolescents score according to physical literacy status. Data are expressed as estimated marginal means (bars) and bias-corrected and accelerated bootstrapped 95% confidence intervals (lines). Correction of multiple comparisons was performed using the false discovery rate *p*-value method developed by Benjamini and Hochberg [22]. Adjusted for sex, age, socioeconomic status, body mass index, energy intake, overall sleep duration, physical activity, and sedentary behavior. KIDMED, Mediterranean Diet Quality Index in children and adolescents; PPL, perceived physical literacy. The KIDMED ranges from −4 to 12 points.

**Figure 4 nutrients-15-04400-f004:**
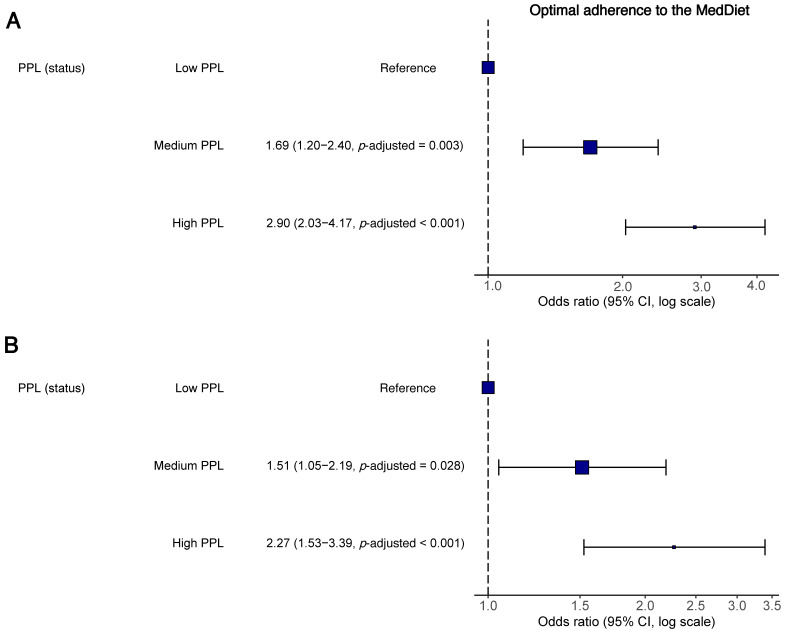
Odds ratio of having an optimal adherence to the Mediterranean diet in relation to perceived physical literacy status among adolescents. Data are presented as dots (odds ratio) and lines (95% confidence intervals). Correction of multiple comparisons was performed using the false discovery rate *p*-value method developed by Benjamini and Hochberg [22]. (**A**) Unadjusted. (**B**) Adjusted for sex, age, socioeconomic status, body mass index, energy intake, overall sleep duration, physical activity, and sedentary behavior. CI, confidence interval; MedDiet, Mediterranean diet; PPL, perceived physical literacy.

**Table 1 nutrients-15-04400-t001:** Main characteristics of the adolescents examined (*n* = 845).

Variable	Low PPL (9–31 Points)	Medium PPL (32–36 Points)	High PPL (31–45 Points)	*p*-Value
Participants (%)	292 (34.6)	304 (36.0)	249 (29.5)	-
Age (years)	14.0 (2.0)	14.0 (2.0)	14.0 (2.0)	0.071
Sex				
Boys (%)	106 (36.3)	148 (48.7)	124 (49.8)	0.002
Girls (%)	186 (63.7)	156 (51.3)	125 (50.2)	
FAS-III (score)	8.0 (3.0)	8.0 (2.2)	9.0 (3.0)	<0.001
BMI (kg/m^2^)	22.5 (7.0)	21.8 (5.9)	20.9 (5.4)	0.001
Energy intake (kcal)	2635.3 (1499.4)	2577.7 (1501.0)	2555.7 (1463.8)	0.678
Overall sleep duration (min)	492.9 (81.4)	497.1 (69.6)	497.1 (60.0)	0.102
YAP-S physical activity (score)	2.4 (0.9)	2.6 (0.8)	2.9 (0.9)	<0.001
YAP-S sedentary behaviors (score)	2.6 (0.8)	2.6 (0.8)	2.4 (0.8)	<0.001
KIDMED (score)	6.0 (4.0)	7.0 (3.0)	8.0 (3.0)	<0.001
Optimal adherence to the MedDiet (%)	78 (26.7)	116 (38.2)	128 (51.4)	<0.001

Data are reported as the median (interquartile range) or count (percentage). BMI, body mass index; FAS-III, Family Affluence Scale III; KIDMED, Mediterranean Diet Quality Index in children and adolescents; MedDiet, Mediterranean diet; Spanish Perceived Physical Literacy Instrument; YAP-S, Spanish Youth Active Profile. The S-PPLI ranges from 9 to 45 points.

## Data Availability

Not applicable.

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
