# Peer review of "Cross-Sectional Association between Perceived Physical Literacy and Mediterranean Dietary Patterns in Adolescents: The EHDLA Study"

_nutrients, 2023, doi:10.3390/nu15204400_

Round 1
Reviewer 1 Report
The study of Dominguez-Martin et al analyzed the relationship between PPL and adherence to Mediterranean diet in a cohort of Spanish adolescents. Findings from the study are in favor of a role of PPL on determining dietary habits among adolescents. The topic is novel and deserve investigation. The methodology used is adequate, the manuscript is well written.
I have only minor concerns
Pag.2 lines 73-76 “However, physical activity…” Please rephrase this sentence
Pag.2 line 78 correct the typo
Take care to spell out all the acronym the first time they are used (i.e. PPL pag. 2 line 78)
Figure 2 should be produced again, as words and number appear to be changed in not decoded symbols
Could the author clarify the main determinants of physical literacy in adolescents? Are there evidence supporting the role of school, youth network, family, social status in providing physical literacy in this population?
Please, take care to correct typos througout the text
Author Response
Reviewer 1
The study of Dominguez-Martin et al analyzed the relationship between PPL and adherence to Mediterranean diet in a cohort of Spanish adolescents. Findings from the study are in favor of a role of PPL on determining dietary habits among adolescents. The topic is novel and deserve investigation. The methodology used is adequate, the manuscript is well written.
Thank you for your valuable time and feedback.
I have only minor concerns
Pag.2 lines 73-76 “However, physical activity…” Please rephrase this sentence
Done. Thank you.
Pag.2 line 78 correct the typo
Done. Thank you.
Take care to spell out all the acronym the first time they are used (i.e. PPL pag. 2 line 78)
Done. Thank you.
Figure 2 should be produced again, as words and number appear to be changed in not decoded symbols
It has been replaced. Thank you.
Could the author clarify the main determinants of physical literacy in adolescents? Are there evidence supporting the role of school, youth network, family, social status in providing physical literacy in this population?
Thank you for your indication. To date, given the recent nature of physical literacy, there is no solid evidence on the determinants of physical literacy in adolescence. However, physical literacy is a determinant of physical activity. Given the close relationship of physical literacy with physical activity, it is possible that they share many common determinants. Since in this study we treated physical literacy as a determinant of adherence to the Mediterranean Diet, we have decided not to increase the information on this aspect.
Reviewer 2 Report
An interesting article titled “Cross-sectional association between perceived physical literacy and Mediterranean dietary patterns in adolescents: The EHDLA study” - introducing new elements to science, with a well-prepared justification of the subject and an interesting discussion.
Nevertheless, it should be suggested:
- ¾ removing teenagers or youths from the keywords - due to repetition of words
¾ it is necessary to explain how the data for calculating BMI were obtained
¾ it should be emphasized whether all questionnaires were validated
¾ whether group size was calculated for the results
¾ whether the data had parametric distributions, as means and standard deviations were calculated
¾ Table 1 is missing statistics – p-values
¾ Figures 1 and 3 are unreadable to the reviewer in the received file
Author Response
Reviewer 2
An interesting article titled “Cross-sectional association between perceived physical literacy and Mediterranean dietary patterns in adolescents: The EHDLA study” - introducing new elements to science, with a well-prepared justification of the subject and an interesting discussion.
Thank you for your valuable time and feedback.
Nevertheless, it should be suggested:
- removing teenagers or youths from the keywords - due to repetition of words
We have decided to keep it since we are within the keyword limit and increase our chances of being found by other researchers.
- it is necessary to explain how the data for calculating BMI were obtained
The next information has been added: “The body weight of the adolescents was measured using an electronic scale (with an accuracy of 0.1 kg) (Tanita BC-545, Tokyo, Japan), while height was determined by a portable height rod with an accuracy of 0.1 cm (Leicester Tanita HR 001, Tokyo, Japan)”. Thank you.
- it should be emphasized whether all questionnaires were validated
This information has been included. Thank you.
- whether group size was calculated for the results
The number of participants for each group have been added.
- whether the data had parametric distributions, as means and standard deviations were calculated
Thank you for your comment. We have replaced the information with medians and interquartile ranges and added information in the statistical analysis section.
- Table 1 is missing statistics – p-values
P-values have been included. Thank you.
- Figures 1 and 3 are unreadable to the reviewer in the received file
Both Figures have been replaced. Thanks.
Reviewer 3 Report
Interesting manuscript analyzing the relationship between perceived physical literacy (PPL) and adherence to the Mediterranean Diet (MedDiet) and its specific components in a sample of Spanish adolescents.
However, Figures 1 and 3 require review as some components are missing or were uploaded with errors, which prevents us from verifying whether the results are consistent with the figures.
Author Response
Reviewer 3
Interesting manuscript analyzing the relationship between perceived physical literacy (PPL) and adherence to the Mediterranean Diet (MedDiet) and its specific components in a sample of Spanish adolescents.
Thank you for your valuable time and feedback.
However, Figures 1 and 3 require review as some components are missing or were uploaded with errors, which prevents us from verifying whether the results are consistent with the figures.
Both Figures have been replaced. Thanks.
Round 2
Reviewer 3 Report
Accepted in the current version, however, it is necessary to improve the quality of figure 1.